# Targeting Wnt/β-Catenin Signaling Exacerbates Ferroptosis and Increases the Efficacy of Melanoma Immunotherapy via the Regulation of MITF

**DOI:** 10.3390/cells11223580

**Published:** 2022-11-12

**Authors:** Hao Wang, Hengxiang Zhang, Yuhan Chen, Huina Wang, Yangzi Tian, Xiuli Yi, Qiong Shi, Tao Zhao, Baolu Zhang, Tianwen Gao, Sen Guo, Chunying Li, Weinan Guo

**Affiliations:** Department of Dermatology, Xijing Hospital, Fourth Military Medical University, Xi’an 710032, China

**Keywords:** Wnt/β-catenin, MITF, ferroptosis, melanoma, anti-PD-1 immunotherapy

## Abstract

Melanoma is the most lethal form of skin cancer, resulting from the malignant transformation of epidermal melanocytes. Recent revolutionary progress in targeted therapy and immunotherapy has prominently improved the treatment outcome, but the survival of melanoma patients remains suboptimal. Ferroptosis is greatly involved in cancer pathogenesis and can execute the outcome of immunotherapy. However, the detailed regulatory mechanisms of melanoma cell ferroptosis remain elusive. Herein, we report that Wnt/β-catenin signaling regulates ferroptosis and melanoma immunotherapy efficacy via the regulation of MITF. First of all, we found that Wnt/β-catenin signaling was prominently suppressed in melanoma cell ferroptosis. Then, we proved that targeting β-catenin exacerbated melanoma cell ferroptosis by promoting the generation of lipid peroxidation both in vitro and in vivo. Subsequent mechanistic studies revealed that MITF mediated the effect of Wnt/β-catenin signaling on melanoma cell ferroptosis, and PGC1α and SCD1 were documented as two main effectors downstream of Wnt/β-catenin-MITF pathway. Ultimately, pharmacological inhibition of β-catenin or MITF increased the efficacy of anti-PD-1 immunotherapy in preclinical xenograft tumor model by promoting ferroptosis. Taken together, Wnt/β-catenin signaling deficiency exacerbates ferroptosis in melanoma via the regulation of MITF. Targeting Wnt/β-catenin-MITF pathway could be a promising strategy to potentiate ferroptosis and increase the efficacy of anti-PD-1 immunotherapy.

## 1. Introduction

The most fatal type of skin cancer is melanoma, originating from the epidermal melanocyte-related malignant transformation in the skin, and its prevalence is steadily rising globally [1]. While rather difficult to cure, current advancements in immunotherapy and targeted therapy have achieved unprecedented advancements in improving the treatment outcome of melanoma [2]. Nevertheless, poor responsiveness and the frequent incidence of treatment confrontation significantly impede the therapeutic effectiveness [2,3,4], and the prognosis of patients with melanoma remains unoptimistic, especially for those with distant metastases [5]. Therefore, elucidating the pathogenesis-related mechanisms for melanoma and establishing additional treatment options might bring some help to melanoma management.

All cancer treatments aim to cause cell death [6]. A cell death paradigm known as ferroptosis is characterized by iron-dependent lipid peroxidation [7], which is significantly involved in melanoma progression and is associated with melanoma immunotherapy. Ferroptosis can be induced in vitro by some specific agents, with RSL3 and ML162 being two potent small-molecule inducers of ferroptosis by targeting GPX4 activity [8,9]. In particular, melanoma cells from lymph nodes are more resistant to ferroptosis, which endows these tumor cells with an increased ability to endure the lymphatic environment and subsequent blood metastasis [10]. In addition, in circulating melanoma cells, the lipogenic regulator SREBP2 triggers the transcription of transferring to suppress ferroptosis and render resistance to targeted therapy, which is correlated with adverse clinical outcomes [11]. More importantly, according to two of the latest reports, tumor cell ferroptosis elicited by IFN-γ secreted from infiltrating CD8^+^ T-cells in the tumor microenvironment improves the efficacy of anti-PD-1 immunotherapy [12,13]. These investigations point out the great clinical significance of intervening ferroptosis in restraining melanoma progression and increasing the effectiveness of immunotherapy. Therefore, some previous investigations have given some insights into the upstream regulatory mechanism of ferroptosis in melanoma. For example, in melanoma cells, miR-137 can adversely control ferroptosis by targeting the glutamine transporter SLC1A5 in a direct manner [14]. In addition, the discovery that miR-9 targets glutamic-oxaloacetic transaminase GOT1 to suppress ferroptosis also supports the vital importance of miRNAs in the control of melanoma cell ferroptosis [15]. Additionally, Nedd4 expression is induced following erastin treatment and ubiquitinates VDAC2/3 to suppress ferroptosis in melanoma, acting as a negative feedback loop to mediate the treatment resistance [16]. While the above reports have brought some insights into ferroptosis regulation, the underlying molecular mechanisms remain far from understood in melanoma.

Wnt/β-catenin signaling is crucial for the development of melanocytes and melanoma pathogenesis, not only guiding the neural crest cells toward differentiation and becoming terminal melanocytes [17,18,19], but also transcriptionally activating downstream targets, such as MITF and driving the transformation and proliferation of melanoma cells [20,21]. Apart from this, Wnt/β-catenin signaling is also greatly implicated in the execution of different paradigms of cell death. In particular, lymphoid enhancer-binding factor 1 (LEF1), Wnt/β-catenin pathway downstream effector, acts as a suppressor of necroptosis via the transcriptional suppression of CYLD in chronic lymphocytic leukemia [22]. Additionally, the Wnt/β-catenin pathway mediates the suppressive influence of berberine on pyroptosis via the regulation of the miR-103a-3p/BRD4 axis [23]. Additionally, the inhibition of canonical Wnt signaling by DKK1 treatment can robustly mitigate PANoptosis (including pyroptosis, apoptosis and necroptosis) in the retinal tissues of streptozotocin-challenged rats with diabetic retinopathy [24]. The crosstalk among Wnt/β-catenin signaling and multiple types of cell death indicates that targeting Wnt might be exploited as a promising strategy to treat cancer and tissue injuries. Recently, it has been reported that platinum-tolerant ovarian cancer cells defined by high Wnt receptor Frizzled-7 expression are more sensitive to ferroptosis inducers, and the transcriptional axis Frizzled-7-β-catenin-Tp63-GPX4 pathway is responsible for this influence [25], suggesting the close relationship between Wnt/β-catenin signaling and ferroptosis. However, the function of Wnt/β-catenin signaling in melanoma cell ferroptosis and the underlying mechanism is ill-defined.

We first discovered in the current research that Wnt/β-catenin signaling was ameliorated in melanoma cell ferroptosis. Following that, we established that the inhibition of the Wnt/β-catenin axis prominently augmented cell ferroptosis in vitro and delayed tumor growth in response to ferroptosis inducer treatment in vivo. A panel of biochemical assays and RNA-sequencing revealed that MITF was responsible for the protective influence of Wnt/β-catenin signaling on melanoma cell ferroptosis, with downstream effectors PGC1α and SCD1 greatly implicated. Finally, whether the synergistic effect of Wnt inhibition/MITF inhibition and anti-PD-1 antibodies on melanoma advancement was dependent on ferroptosis was also examined in vivo. 

## 2. Materials and Methods

### 2.1. Cell Culture and Reagents

Human melanoma cell lines A2058, A375 and mouse melanoma cell line B16F10 were grown in DMEM (cat. 12100061, Gibco, Scotland, UK) containing 10% (*v*/*v*) FBS (Procell, Wuhan, China), 2 mM L-glutamine (Invitrogen, Scotland, UK) and 1% (*v*/*v*) penicillin/streptomycin (V900929100, Sigma-Aldrich, St. Loius, MO, USA). An environment containing 5% CO_2_ (37 °C) was used to grow the cells. In 2016, all cell lines were verified to be mycoplasma-free by short tandem repeat fingerprinting at the Fourth Military Medical University’s DNA typing center. Laduviglusib (CHIR-99021), HCl (C91) (cat. S2924), Mito-TEMPO (cat. S9733), liproxstatin-1 (cat. S7699) and lithium chloride (LiCl) (cat. E0153) were purchased from Selleck (Houston, TX, USA). RSL3 (cat. HY-100218A), ICG001 (cat. HY-14428), necrosulfonamide (cat. HY-100573), nelfinavir mesylate (cat. HY-15287A), ferrostatin-1 (cat. HY-100579), A939572 (cat. HY-50709) and Z-VAD-FMK (cat. HY-16658B) were purchased from MedChemExpress (Princeton, NJ, USA). Anti-PD-1 antibody (cat. BE0146) was obtained from Bio X Cell (West Lebanon, NH, USA).

### 2.2. Cell Viability Assay and Colony Formation

For the cell counting kit-8 (CCK-8) (GK10001, GLPBIO, Montclair, CA, USA) assay, inoculation of A2058 and A375 cells was completed in a 96-well plate with a density of 5000 or 6000 cells/well and was stimulated with or without the indicated reagents. Following that, the fresh medium comprising 1/10 CCK-8 reagent was added for incubation for 1 h at 37 °C with 5% CO_2_, and a microplate reader (Model 680, Bio-Rad, Hercules, CA, USA) was employed to determine the absorbance of each well at 450 nm. To perform the colony formation assay, cells were digested into single-cell suspension, and on 6-well plates, 2000 cells/well were cultured under adherent environments in full media to facilitate colony formation and incubated for 14 days until obvious cell colonies appeared. Then, the culture media were removed, and the colonies were rinsed with PBS, stabilized in 5% formaldehyde for 15 min, stained with crystal violet staining solution for 25 min, flushed with ddH_2_O and dried at room temperature. The fraction of survival was computed using the number of colonies formed and averaged from the duplicate wells. 

### 2.3. qRT-PCR Analysis

Employing the TRIzol reagent (cat. 15596018, Invitrogen, Carlsbad, CA, USA), total RNA was extracted. With the aid of the PrimeScript RT Master Mix kit (cat. RR036A, TaKaRa, Kyoto, Japan), the isolated RNA was reverse-transcribed to cDNA based on the methods described in the guidelines of the manufacturer. Subsequent qRT-PCR analysis was conducted employing the SYBR Premix Ex Taq II kit (cat. RR820A, TaKaRa, Kyoto, Japan) with a multicolor real-time PCR detection system (Bio-Rad iQTM5, Hercules, CA, USA). Appendix A depicts the primer sequences utilized in this research for PCR analysis. The quantification was investigated using the 2^−ΔΔCT^ technique, and the gene expression levels were all normalized to β-actin mRNA [8,9].

### 2.4. RNA Sequencing

Extraction of total RNA was conducted with Trizol reagent from A2058 melanoma cells that received the treatment with 2 μM RSL3 or RSL3 plus 6 μM ICG001 for 24 h. Subsequent RNA sample processing and RNA sequencing analysis services were all provided by Gene Denovo Biotechnology Co using Illumina Novaseq6000 (Guangzhou, China).

### 2.5. RNA Interference and Gene Transfection

From GenePharma (Shanghai, China), small interfering RNAs were procured. The sequences of β-catenin small interference RNAs are the following: si-β-catenin-1: 5′-GCTTGGAATGAGACTGCTGAT-3′; si-β-catenin-2: 5′-TTGTTATCAGAGGACTAAATA-3′. In accordance with the manufacturer’s protocol, utilizing Lipofectamine 3000 (cat. L3000-015, Invitrogen, Carlsbad, CA, USA) reagent, transfection of A2058 and A375 cells was performed with these small interference RNAs.

### 2.6. Antibodies

The primary antibodies used in this investigation are listed below, along with the dilutions for immunofluorescence (IF) and western blotting (WB) analysis staining analysis: β-catenin (cat. 8480, CST, Danvers, Massachusetts, USA, 1:1000 for WB), active β-catenin (cat. 8814, CST, 1:200 for IF, 1:1000 for WB), MITF (cat. ab12039, Abcam, Cambridge, UK, 1:1000 for WB), PGC1α (cat. 66369-1-Ig, Proteintech, Wuhan, China, 1:1000 for WB), SCD1 (cat. 28678-1-AP, Proteintech, 1:1000 for WB), Ki67 (cat. 9449, CST, Boston, MA, USA, 1:400 for IF) and PTGS2 (cat. 12282, CST, 1:800 for IF). Secondary antibodies for WB were horseradish peroxidase-conjugated goat antirabbit or anti-mouse IgG (cat. 7074P2, CST, 1:3000). Secondary antibodies for IF were goat anti-mouse IgG-Alexa Fluor 488 conjugated (zhuangzhiBIO, Xi’an, China, 1:200) and goat antirabbit IgG-Cy3 conjugated (zhuangzhiBIO, 1:200).

### 2.7. Lipid ROS Production Measurement

Onto 12-well plates, A2058 and A375 were plated at a density of 1.2 × 10^5^ per well and cultured for 24 h to ensure cell adhesion. One mL fresh medium mixed with the following stimulations was then utilized to incubate the cells: RSL3, 2 μM for A2058 and 0.5 μM for A375; 6 μM ICG001; 5 μM Nelfinavir; 5 μM C91; 2.5 μM A939572; 10 μM Mito-tempo and 10 μM Ferrostatin-1 for both A2058 and A375. After 24 h, these cells were co-cultured with 1 mL of fresh DMEM comprising 5 μM C11-BODIPY 581/591 (cat. GC40165, GLPBIO, Montclair, CA, USA) at 37 °C for 30 min. Subsequently, cells were subjected to washing twice with PBS, harvested with trypsinization, washed again and kept in resuspension of 400 μL of PBS buffer containing 5% FBS. The cell suspension was analyzed on a flow cytometer to determine the amount of intracellular ROS [8,9]. The maximum emission wavelength of oxidized BODIPY shifted from 591 nm to 510 nm, which indirectly reflected the level of lipid peroxide. The lipid ROS levels were examined by employing the BD LSRFortessa Flow Cytometer (BD Biosciences, Franklin Lake, NJ, USA). 

### 2.8. Malondialdehyde (MDA) Assay

Utilizing an MDA assay kit (cat. A003-1-2, Nanjing Jiancheng, Nanjing, China), the relative MDA concentration in cell lysates was detected in accordance with the manufacturer’s guidelines. In brief, T75 flasks were used to seed melanoma cells (6 × 10^6^ cells per flask) using stimulations related to Lipid ROS production measurements (2.7). Subsequent cells were trypsinized (cat. PB180228) and counted and MDA then reacted with thiobarbituric acid (TBA) to produce an MDA-TBA adduct at 95 °C under an acidic condition. At 532 nm, adduct absorbance was measured employing a Microplate Reader (Model 680, Bio-Rad, Hercules, CA, USA).

### 2.9. Mitochondrial ROS (mitoROS) Detection

A2058 and A375 were cocultured with the following stimulations: RSL3, 2 μM for A2058 and 0.5 μM for A375; 6 μM ICG001 and 10 μM Mito-tempo for both A2058 and A375. In order to detect mitoROS, cells were harvested, washed once with DMEM and suspended in 5 μM MitoSOX (M36008, Invitrogen, Carlsbad, CA, USA) at 37 °C for 15 min, keeping them out of the light. With prewarmed PBS at 37 °C, the cells were then washed twice and resuspended in prewarmed PBS for detection by the BD LSRFortessa Flow Cytometer.

### 2.10. Western Blotting Analysis

After indicated stimulations, cells were rinsed two times with precooled PBS buffer and supplemented lysed in RIPA buffer (P0013B, Beyotime, Shanghai, China) with protease inhibitor cocktail (cat. 5871, CST). Cell lysates were incubated for 5 min on ice and clarified via centrifugation at 12,000× *g* for 10 min. Then, the bicinchoninic acid (BCA) Assay Kit (P0012, Beyotime) was employed to investigate the protein concentrations. Total protein samples were mixed with 5 × SDS-PAGE loading buffer (P0015, Beyotime) (4:1) and denatured at 95 °C for at least 10 min. Protein samples (16–25 μg) were subsequently separated using SDS-PAGE (P0012A, Beyotime) and placed onto PVDF membranes (IPVH00010, Millipore Sigma, Burlington, MA, USA). Then, the blockage of membranes was performed for 1.5 h in 5–6% *w*/*v* nonfat milk and incubated with primary antibodies at 4 °C for 18 h. After three TBST washes, the membranes were treated with HRP-conjugated secondary antibodies (CST) at room temperature for 1 h. Signal was detected utilizing ECL Western Blotting Substrate (Tanon, Shanghai, China), and the images were captured and analyzed employing the ChemiDoc XRS^+^ System (Bio-Rad, Hercules, CA, USA).

### 2.11. IF Staining Analysis

Melanoma cells A2058 and A375 were grown in 15 mm glass bottom cell culture dishes (NEST, Wuxi, China), washed two times with PBS and stabilized with 4% paraformaldehyde for 15 min at room temperature. Then, utilizing 0.5% of Triton X-100 for 10 min, cells were permeabilized, washed with PBS and blocked for 1 h with 10% goat serum (Gibco, Carlsbad, CA, USA) to block nonspecific sites of antibody adsorption. At 4 °C, following overnight incubation with the primary antibody, cells were washed thrice for 5 min with PBS and then were treated with fluorochrome-conjugated secondary antibodies in a dark chamber for 1 h. Finally, cells were washed thrice and incubated for 10 min with FlucroshieldTM with DAPI (Sigma-Aldrich, St. Louis, MO, USA). The stained cells were observed using a Zeiss LSM880 confocal laser scanning microscope (Zeiss, Oberkochen, Germany).

### 2.12. Measurement of ATP Level and Luciferase Assay

In 6-well plates, melanoma cells A2058 and A375 were seeded at a density of 2 × 10^5^ per well for 24 h. Cells were collected and lysed after following treatment for 24 h (RSL3, 2 μM for A2058 and 0.5 μM for A375; 6 μM ICG001 for both A2058 and A375). Then, the ATP level was investigated using a mixture of 20 μL cell supernatant with 100 μL luciferase reagent employing an ATP Assay Kit (S0026, Beyotime, Shanghai, China). Luciferase activity, quantified by the ATP level, was normalized to total protein and analyzed utilizing the BCA Assay Kit. For the canonical Wnt signaling activity luciferase assay, the SuperTOPFlash (D2505, Beyotime) was transfected and analyzed using a luciferase reporter gene detection kit (RG005, Beyotime) using a luminometer (Promega GloMax 20/20, Singapore) according to the manufacturer’s instructions.

### 2.13. Fe^2+^ Assay

Cells were treated with the following stimulations for 24 h: RSL3, 2 μM for A2058 and 0.5 μM for A375; 6 μM ICG001 for both A2058 and A375. The Fe^2+^ content in these cells was examined employing the Ferrous iron Colorimetric Assay Kit (E-BC-K773-M, elabscience, Wuhan, China) in accordance with the manufacturer’s guidelines. Spectrophotometry was adopted to measure the absorbance at a wavelength of 593 nm.

### 2.14. Cell Treatments

Firstly, to explore the ability of RSL3 and ML162 to induce ferroptosis using a cell viability assay, (0.125–8 μM) RSL3 or (1–16 μM) ML162-stimulated A2058 and A375 melanoma cells were pretreated with 10 μM ferrostatin-1, 10 μM Z-VAD-FMK or 0.5 μM necrosulfonamide for 24 h. For western blotting, IF, Topflash and qRT-PCR assays, melanoma cells were treated with RSL3 or ML162 for 24 h (RSL3, 2 μM for A2058 and 0.5 μM for A375; ML162, 4 μM for both A2058 and A375). Moreover, to investigate the effect of Wnt/β-catenin signaling on melanoma cell ferroptosis, as well as the underlying mechanism, A2058 and A375 melanoma cells were treated with the following agents of indicated concentrations for 24 h (RSL3, 2 μM for A2058 and 0.5 μM for A375; 6 μM ICG001; 5 μM nelfinavir; 5 μM C91; 2.5 μM A939572; 10 μM Mito-tempo and 10 μM Ferrostatin-1 for both A2058 and A375). 

### 2.15. Xenograft Mice Model and Treatments

After 1-week acclimatization, six-week-old female nude mice (18–22 g) were selected and randomly divided into six groups. A375 cells (5 × 10^6^ cells in 200 μL of 1 × PBS) were injected subcutaneously into the right posterior side of the mice to generate subcutaneous tumors. Then, the tumor growth of mice was monitored by quantifying tumor length (L) and width (W) (tumor volume = L × W^2^/2). Liproxstatin-1 (10 mg/kg), ICG001 (20 mg/kg), or RSL3 (25 mg/kg) were injected intraperitoneally in nude mice, either alone or in combination, for 13 days [26,27,28]. The mice were sacrificed after the indicated treatment for 13 days. Tumor tissues were then harvested, weighed and photographed. For IF staining analysis, tumors were stabilized in 4% paraformaldehyde overnight and subjected to embedding in paraffin. 

To investigate the function of Wnt/β-catenin-MITF signaling in melanoma immunotherapy, six-week-old female C57BL/6 mice (18–22 g) were applied to establish xenograft tumor model via the injection of B16F10 cells (5 × 10^5^ cells in 100 μL of 1 × PBS). C57BL/6 mice were intraperitoneally injected with liproxstatin-1 (10 mg/kg), ICG001 (20 mg/kg), nelfinavir (100 mg/kg) or mouse anti-PD1 antibodies unaccompanied or in combination for 10 days [26,27,28,29]. Then the tumor tissues were collected and analyzed using the same methods described above.

### 2.16. Statistical Analysis

At least three independent replications of each experiment were conducted. The data are expressed as the standard error of the mean. Statistical levels (*p*-values) were calculated utilizing unpaired Student’s *t*-tests with GraphPad Prism 8.0. * *p* < 0.05; ** *p* < 0.01; *** *p* < 0.001; ns, nonsignificant.

## 3. Results

### 3.1. Wnt/β-Catenin Signaling Is Prominently Suppressed in Melanoma Cell Ferroptosis

RSL3 and ML162 are two potent small molecules that induce ferroptosis by targeting GPX4 activity [8,9]. We first investigated the ability of these two agents to induce ferroptosis. It was discovered that RSL3 and ML162 could cause cell death in A2058 and A375 melanoma cell lines in a manner dependent on dose, which could be reversed by the ferroptosis-specific inhibitor ferrostatin-1 but not reversed by the apoptosis inhibitor Z-VAD-FMK or necroptosis inhibitor necrosulfonamide (Figure 1a). Through immunoblotting analysis, while the expression of total β-catenin was unaltered, the level of active β-catenin was substantially downregulated in response to either the treatment of RSL3 or ML162 (Figure 1b). The IF staining analysis also displayed that the intensity of active β-catenin was significantly suppressed in response to ferroptosis inducer treatment in both melanoma cell lines (Figure 1c). Similarly, the qRT-PCR analysis showed that the canonical downstream targets of β-catenin, including Axin2, c-Myc and CyclinD1, were significantly reduced (Figure 1d). To confirm the impairment of β-catenin activation in melanoma ferroptosis, Topflash reporter assay was employed to detect the transcriptional activity of β-catenin upon the treatment with RSL3 or ML162. Of note, while the positive control LiCl induced robust upregulation of reporter signaling as previously reported [9], RSL3 or ML162 treatment was capable of significantly suppressing the reporter signaling compared with control in the melanoma cell lines A2058 and A375 (Figure 1e). Taken together, the Wnt/β-catenin signaling activation was substantially impaired in melanoma cell ferroptosis.

### 3.2. Targeting Wnt/β-Catenin Signaling Exacerbates Melanoma Cell Ferroptosis by Promoting the Production of Lipid Peroxidation

The next step was to investigate if Wnt/β-catenin signaling modulates ferroptosis. First of all, we tested the inhibitory influence of the pharmacological inhibitor ICG001 at different concentrations on the cell viability of A2058 and A375 melanoma cells and found that ICG001 at a concentration of more than 6 μM was capable of inducing prominent suppression of cell viability (Appendix A). Meanwhile, ICG001 at a concentration of 6 μM could suppress the expression of active-β-catenin in melanoma cells, where it had no impact on the total β-catenin expression (Figure 2a). As revealed by the CCK8 assay, though ICG001 monotreatment showed marginal influence, RSL3 was capable of inducing prominent suppression of colony formation and cell viability (Figure 2b,c). Moreover, the combination of ICG001 and RSL3 could induce additional attenuation of the viability of cells and the formation of a colony in the melanoma cell lines A2058 and A375, which could be almost reserved by ferrostatin-1 (Figure 2b,c). One unique characteristic of ferroptosis is the production of excessive lipid peroxidation [30], and one of lipid peroxidation’s most significant end products is MDA. Therefore, we wondered whether Wnt/β-catenin signaling affects ferroptosis via the regulation of MDA. As shown, RSL3 treatment could significantly enhance intracellular MDA, whereas ICG001 cotreatment could potentiate MDA accumulation (Figure 2d). In line with this, lipid ROS identified through flow cytometry with the employment of the C11-BODIPY probe also exhibited that the combined administration of ICG001 and RSL3 could facilitate the generation of lipid ROS and thereby lipid peroxidation (Figure 2e). Apart from the pharmacological inhibition of β-catenin, we also employed genetic knockdown of β-catenin to examine the influence of β-catenin on melanoma cell ferroptosis (Figure 2f). In light of what CCK8 revealed, the knockdown of β-catenin expression could potentiate RSL3-induced suppression of cell viability, and ferrostatin-1 may be able to reverse this phenomenon (Figure 2g). Meanwhile, colony formation assays also revealed that β-catenin expression deficiency exacerbated RSL3-induced ferroptosis, and ferrostatin-1 may also be able to reverse this event (Figure 2h). In consistency with this, β-catenin expression deficiency facilitated the generation of MDA and lipid ROS in response to RSL3 administration in the A2058 and A375 melanoma cell lines (Figure 2i,j). Therefore, inhibiting Wnt/β-catenin signaling could exacerbate RSL3-induced cell death in a manner dependent on ferroptosis via the facilitation of lipid peroxidation generation. 

To confirm the inhibitory influence of targeting Wnt/β-catenin signaling on ferroptosis in vivo, a preclinical transplantation mouse model was created by subcutaneously injecting A375 melanoma cells into nude mice (Figure 3a). After growth for 12 days, RSL3 was intraperitoneally injected unaccompanied or in combination with ICG001 for the subsequent 13 days (Figure 3b). The RSL3 monotreatment resulted in a prominent delay in tumor growth, and ICG001 coadministration could suppress the progression of the tumor, as revealed by tumor weights and tumor volumes (Figure 3c,d). This combined influence could be prominently reversed by the coadministration of the ferroptosis-specific inhibitor liproxtatin-1(Figure 3c,d). The expression of PTGS2, a marker of ferroptosis [31], was enhanced upon RSL3 treatment and further enhanced after ICG001 cotreatment, according to isolated tumor-related IF staining results (Figure 3e). In addition, the staining intensity of Ki67 in the transplanted tumor was also prominently suppressed after the combined treatment with both RSL3 and ICG001, which could be reversed by the cotreatment with liproxtatin-1 (Figure 3e). Together, the inhibition of Wnt/β-catenin signaling may potentially improve the effectiveness of the ferroptosis inducer in vivo. 

It should be noted that the excessive load of intracellular ferrous iron (Fe^2+^) is another hallmark characteristic according to the previous report [32], which provides the molecular basis to protect cells against ferroptosis via iron chelator agents, such as deferoxamine [33]. In comparison with the control group, monotreatment with RSL3 could induce a significant increase in intracellular Fe^2+^ levels (Appendix A). However, cotreatment with ICG001 failed to potentiate this increasing trend (Appendix A), indicating that the facilitative influence of Wnt signaling deficiency on ferroptosis is not related to an excessive load of Fe^2+^. Taken together, targeting Wnt/β-catenin signaling exacerbates melanoma cell ferroptosis by enhancing lipid peroxidation production, instead of affecting the excess Fe^2+^. 

### 3.3. MITF Mediates the Influence of Wnt/β-Catenin Signaling on Melanoma Cell Ferroptosis

To investigate the mechanism underlying the function of Wnt/β-catenin signaling in melanoma cell ferroptosis, RNA sequencing was employed to test the differentially expressed molecules between the RSL3 monotreatment group and the RSL3 plus ICG001 combined treatment group in the A2058 melanoma cells. According to the results, there were a total of 1153 molecules displaying more than a 2-fold change in the combined treatment group in comparison with the monotreatment group (Appendix A, Figure 4a). The gene set enrichment analysis (GSEA) discovered that the differentially expressed molecules were enriched in the biosynthesis of unsaturated fatty acids, oxidative phosphorylation, fatty acid elongation, cysteine and methionine metabolism and ferroptosis pathways that are tightly associated with ferroptosis (Figure 4b). To be specific, the expression of the canonical melanocytic lineage-specific transcriptional factor MITF, which is also the downstream target of Wnt/β-catenin signaling, was significantly reduced, so were its downstream transcriptional targets, including TYR, DCT, PPARGC1A, TRPM1, BCL2A1, SCD, RAB27A and MLANA [34,35] (Figure 4c). MITF has been regarded as one of the most important melanocytic lineage-specific transcriptional factors driving melanocyte differentiation and melanoma development [36], and concurrent qRT-PCR and immunoblotting analysis revealed that either the protein or mRNA levels of MITF was prominently downregulated in the RSL3+ICG001 combined treatment group, compared with a monotreatment group (Figure 4d). In addition, the mRNA levels of TYR, TYRP1 and DCT were also downregulated in the combined treatment group (Figure 4d). The observation that Wnt/β-catenin signaling controls MITF expression and its downstream targets prompted us to see whether MITF facilitates the function of Wnt/β-catenin signaling in ferroptosis, which may raise the notion that the modulatory role of Wnt/β-catenin signaling in ferroptosis is lineage-specific. Nelfinavir is an HIV-1-portease inhibitor proven to suppress any influence on MITF expression (Figure 4e) [29]. We found that nelfinavir treatment at the concentration of nonlethal influence could exacerbate RSL3-induced suppression of cell viability that might be effectively reversed via coadministration with ferrostatin-1 (Figure 4f), which mimics the influence of targeting Wnt in sensitizing melanoma cells to RSL3-induced cell ferroptosis (Figure 2b). In line with this, lipid ROS assayed utilizing flow cytometry with the employment of a C11-BODIPY probe revealed that nelfinavir treatment could promote the generation of lipid ROS in response to RSL3 treatment, and ferrostatin-1 may be able to reverse this phenomenon (Figure 4g). 

Thereafter, we tested whether MITF mediated the functioning of Wnt/β-catenin signaling in melanoma cell ferroptosis. The immunoblotting analysis revealed that Wnt agonist C91 could induce prominent upregulation of active β-catenin, and in parallel, the expression level of MITF was also significantly increased (Figure 4h). In addition, while the RSL3 treatment induced the downregulation of both active β-catenin and MITF, cotreatment with C91 was capable of reversing this alteration trend (Figure 4h). While Wnt agonist C91 treatment significantly reversed the influence of ferroptosis induction by RSL3 in both A2058 and A375 cell lines, cotreatment with nelfinavir could abolish the protective influence of Wnt activation, as revealed by the CCK8 assay (Figure 4i). In line with this, nelfinavir could reverse the influence of C91 on suppressing the generation of lipid ROS in response to RSL3 treatment (Figure 4j). In total, these data demonstrate that MITF mediates the influence of Wnt/β-catenin signaling on ferroptosis in melanoma.

### 3.4. PGC1α and SCD1 Are Two Main Effectors Downstream of Wnt/β-Catenin-MITF Pathway in Melanoma Cell Ferroptosis

Next, we went on to figure out the downstream effectors that mediated the functioning of the Wnt/β-catenin-MITF pathway in ferroptosis. The findings obtained from RNA sequencing data revealed that PGC1α (encoded by *PPARGC1A*) and SCD1 (encoded by *SCD1*) were downregulated in response to the combined treatment with ICG001 and RSL3, compared with the RSL3 monotreatment group (Figure 4c). For confirmation, qRT-PCR and immunoblotting were conducted simultaneously, and the protein and mRNA levels of PGC1α and SCD1 were prominently reduced in the combined treatment group compared with the RSL3 monotreatment group (Figure 5a,b). PGC1α is the master regulator of mitochondrial function and regulates the mitochondrial antioxidant defense system via the downstream antioxidant molecules, including superoxide dismutase 2 (SOD2), catalase, thioredoxin 2 (TRX2), thioredoxin reductase 2 (TXNRD2), peroxiredoxin 5 (Prx5) and Prx3 [37,38]. More importantly, previous reports have demonstrated that mitochondrial dysregulation is greatly implicated in ferroptosis induction [39,40,41]. Therefore, we hypothesized that the deficiency of PGC1α expression might impact the mitochondrial antioxidant system and functional status of participating in the regulation of ferroptosis. As was revealed, the transcriptional levels of SOD2, catalase, TRX2, TXNRD2, Prx5 and Prx3 that were downstream targets of PGC1α were ubiquitously downregulated in response to the combination of RSL3 and ICG001, compared with the RSL3 monotreatment group (Figure 5c). In addition, the mRNA level of COXIV, which reflects the number of mitochondria, was also downregulated in the combined treatment group (Figure 5d). In consistency with these results, the level of intracellular ATP was prominently decreased in response to the combination of RSL3 and ICG001, compared with the RSL3 monotreatment group (Figure 5e). In addition, the level of mitoROS was also substantially decreased in the combined treatment group compared to other groups (Figure 5f). Then, we used Mito-TEMPO, which is a mitochondria-targeted superoxide dismutase mimetic with superoxide and alkyl radical scavenging properties, to scavenge mitoROS (Figure 5f) [42], intending to see whether the induction of ferroptosis upon the combination of RSL3 and ICG001 was dependent on mitoROS. As was revealed, Mito-TEMPO pretreatment could ameliorate the attenuation of cell viability triggered by RSL3 plus ICG001 in melanoma cells (Figure 5g). Additionally, Mito-TEMPO treatment was able to reduce the generation of lipid ROS in response to the combined treatment (Figure 5h). In total, the above results demonstrated that PGC1α mediates the execution of ferroptosis downstream of MITF via the regulation of the mitochondrial status and antioxidant system.

Previous reports have demonstrated that SCD1 is greatly implicated in protecting cancer cells against the induction of ferroptosis [43]. Since we found that the expression of SCD1 was also reduced in response to the group receiving combined treatment in comparison with the RSL3 monotreatment group (Figure 5a,b), and SCD1 was an important downstream target of MITF [44], we suggested that SCD1 could function downstream of MITF in the regulation of ferroptosis. Pharmacological inhibition of SCD1 by A939572 was employed in melanoma cells undergoing treatment with RSL3. In light of what CCK8 revealed, the suppression of SCD1 activity could exacerbate RSL3-induced melanoma cell ferroptosis, and ferrostatin-1 may be able to reverse this phenomenon (Figure 5i). In line with this, SCD1 inhibition could promote the generation of lipid ROS in melanoma cells undergoing RSL3-induced ferroptosis, and ferrostatin-1 might also be able to reverse this phenomenon (Figure 5j). More importantly, pharmacological inhibition of SCD1 significantly abrogated the protective influence of Wnt agonists on melanoma cell ferroptosis, as revealed by both cell viability and lipid ROS results obtained from the CCK8 assay and C11-BODIPY probe-based flow cytometry (Figure 5i,j). Therefore, these facts demonstrate that SCD1 also participates in the regulation of ferroptosis downstream of MITF. 

### 3.5. Targeting Wnt/β-Catenin and MITF Increases the Effectiveness of Anti-PD-1 Antibodies via the Induction of Ferroptosis

Treatment of melanoma with anti-PD-1 immunotherapy is significantly effective, while poor responsiveness and resistance to the treatment considerably reduce its effectiveness [45]. A previous report has revealed that the melanoma cell-intrinsic Wnt/β-catenin signaling activation is associated with the nonexistence of a T-cell gene expression signature, rendering the resistance to anti-PD-L1/anti-CTLA-4 monoclonal antibody treatment [46]. Moreover, pharmacological Wnt ligand inhibition promotes anti-PD-1 effectiveness via reversing dendritic cell tolerization and the recruitment of granulocytic myeloid-derived suppressor cells in tumor models [47]. These reports have mainly emphasized the regulatory influence of tumorous Wnt/β-catenin signaling on surrounding immune cells in the tumor microenvironment (TME). Since that IFN-γ derived from activated CD8^+^ T-cells in TME can trigger tumor cell ferroptosis upon anti-PD-1 antibody treatment, and targeting Wnt/β-catenin signaling can exacerbate ferroptosis according to our results, we proposed that targeting Wnt/β-catenin might increase the effectiveness of anti-PD-1 antibody by the induction of tumor cell ferroptosis. To produce a xenograft tumor model, B16F10 mouse melanoma cells were injected subcutaneously into immune-competent C57BL/6 mice (Figure 6a). As shown, anti-PD-1 antibody treatment resulted in slight tumor regression, which was forwardly suppressed after the coadministration with Wnt inhibitor ICG001 (Figure 6b–d). This combined influence was abolished by the ferroptosis inhibitor liproxstatin-1 treatment, indicating that targeting Wnt/β-catenin can enhance the effectiveness of anti-PD-1 antibody through the ferroptosis induction (Figure 6b–d). In isolated tumors, simultaneous IF staining analysis of PTGS2 discovered that the combination of anti-PD-1 antibody and ICG001 indeed induced more tumor cell ferroptosis, and liproxstatin-1 coadministration could suppress the synergized therapeutic influence (Figure 6e). In parallel, the staining intensity of Ki67 was substantially reduced after the combined treatment with both anti-PD-1 antibodies and ICG001, and liproxstatin-1 may be able to reverse this phenomenon (Figure 6e).

Apart from this, we also employed the B16F10 xenograft tumor model to test whether targeting MITF could enhance the effectiveness of anti-PD-1 antibodies via ferroptosis (Figure 6a). It was revealed that systemic targeting of MITF by nelfinavir led to a prominent delay of tumor growth, and targeting MITF was capable of potentiating the treatment influence of anti-PD-1 antibodies that could be reversed by liproxstatin-1, as was revealed by both tumor weight and tumor volume (Figure 6f–h). In isolated tumors, concurrent IF staining analysis of PTGS2 discovered that the combination of anti-PD-1 antibodies and nelfinavir indeed induced more tumor cell ferroptosis, and the staining intensity of Ki67 was significantly reduced after the combined treatment with both anti-PD-1 antibodies and nelfinavir (Figure 6i). Together, targeting Wnt/β-catenin signaling and MITF can enhance the effectiveness of anti-PD-1 antibodies partially through ferroptosis induction. 

## 4. Discussion

We first discovered in the current research that Wnt/β-catenin signaling was significantly ameliorated in response to the induction of ferroptosis. Then, it was proved that targeting Wnt/β-catenin signaling exacerbated melanoma cell ferroptosis by enhancing lipid peroxidation production. Subsequently, through the employment of RNA sequencing technology and a panel of biochemical assays, melanocytic-specific transcriptional factor MITF was identified to mediate the influence of Wnt/β-catenin signaling in melanoma cell ferroptosis. In particular, PGC1α and SCD1 were proved as two main effectors downstream of the Wnt/β-catenin-MITF pathway in melanoma cell ferroptosis. Ultimately, the results of the preclinical mouse model demonstrated that targeting Wnt/β-catenin or MITF could significantly enhance the effectiveness of anti-PD-1 antibody via ferroptosis induction. Collectively, these findings established that targeting Wnt/β-catenin signaling exacerbates ferroptosis in melanoma via the regulation of MITF (Figure 7). The intervention of Wnt/β-catenin and MITF might be explored as a melanocytic-specific strategy to enhance the effectiveness of immunotherapy in melanoma. 

Iron-dependent lipid peroxidation is a hallmark of a newly found cell death mode known as ferroptosis. Earlier research has demonstrated that ferroptosis participated in the regulation of tumor cell metastasis and the effectiveness of targeted therapy [47,48]. More importantly, the activation of tumor-infiltrating CD8^+^ T-cells in response to anti-PD-1 immunotherapy can promote the secretion of IFN-γ that suppresses the expression of the Xc-system in the tumor, so that to trigger tumor cell ferroptosis [13]. Therefore, the intervention of ferroptosis is promising in restraining melanoma progression and increasing the effectiveness of either targeted therapy or immunotherapy. Some studies have demonstrated the upstream regulatory mechanism of ferroptosis in melanoma before [14,15,16], and we previously reported that CAMKK2 and miR-21-3p were critical regulators of melanoma cell ferroptosis through the regulation of AMPK-Nrf2 and TXNRD1, respectively [8,49]. Herein, we proved that Wnt/β-catenin signaling was also implicated in melanoma cell ferroptosis, extending the upstream regulatory network of ferroptosis in melanoma. A recent report has demonstrated that in gastric cancer, the Wnt/β-catenin signaling activation reduces cellular lipid ROS generation and then impedes ferroptosis. In particular, the β-catenin/TCF4 transcription complex could bind to the GPX4′s promoter region in a direct manner and induce its expression [50]. Different from this investigation, our present study has revealed that MITF mediates the antiferroptosis function of Wnt/β-catenin signaling, indicating the existence of a melanocytic-lineage-specific mechanism responsible for the regulation of ferroptosis by Wnt/β-catenin signaling. Therefore, targeting the Wnt/β-catenin-MITF pathway could be a melanocytic lineage-specific strategy to exacerbate ferroptosis. 

Significant evidence has emerged to demonstrate that MITF coordinates melanocyte and melanoma biology in ways more than just increasing melanocyte cell identity and melanosomal gene regulation [51]. High MITF activity levels are linked to cell differentiation and attenuated proliferation, according to the “rheostat model”. However, steadily declining MITF activity levels are linked to proliferation, dedifferentiation/invasion (as seen in melanoma cells), senescence and ultimately cell death [52]. MITF can transcriptionally regulate the expression of BCL2 to enable melanoma cell survival [53], whereas the function of MITF in alternative novel cell modalities, such as ferroptosis, remains elusive. Distinct downstream target genes are responsible for the versatile functions of MITF in different aspects of melanocyte and melanoma biology, including DNA damage repair, cell metabolism, cell differentiation, invasion and metastasis [51]. Of note, the master regulator of mitochondrial function and antioxidant system, PGC1α, is the main transcriptional target of MITF and regulates mitochondrial biogenesis to promote tumor growth in melanoma [54]. In addition, the key lipogenic enzyme SCD1 is also the target of MITF that mediates the process of fatty acid composition and melanoma phenotype switching [44]. Through the RNA-sequencing technology and a panel of biochemical assays, it was evidenced that MITF was significantly involved in the regulation of melanoma cell ferroptosis downstream of Wnt/β-catenin signaling, and pharmacological suppression of MITF by nelfinavir was effective in exacerbating ferroptosis, which could be considered a promising strategy with relatively high translational potential. Since ferroptosis is a cell death modality characterized by excessive lipid peroxidation, we emphasized more on the effectors that participate in the control of oxidative stress and lipid metabolism downstream of MITF, and PGC1α and SCD1 were documented as two main effectors in melanoma cell ferroptosis. SCD1 has been reported as a ferroptosis suppressor via the regulation of monounsaturated fatty acid synthesis in ovarian cancer cells [43]. Additionally, PGC1α has been proven to attenuate oxidative stress and ferroptosis through the regulation of an Nrf2-dependent antioxidative system in subarachnoid hemorrhage in rats [55]. The results of our present study are consistent with these previous reports and provide solid mechanistic evidence to support the crucial function of MITF in the control of ferroptosis. 

The blockade of PD-1/PD-L1 has significantly improved the prognosis of patients with melanoma [2]. Of note, two studies have highlighted that the induction of tumor cell ferroptosis triggered by IFN-γ resulting from infiltrated CD8^+^ T-cells mediates the influence of melanoma immunotherapy [12,13]. Our previous investigations proved that miR-21-3p and CAMKK2 regulated the TXNRD1 and AMPK-Nrf2 axes, respectively, to mediate the execution of melanoma cell ferroptosis [8,49]. Either nanoparticle delivery of miR-21-3p or systemic administration of a CAMKK2 inhibitor could prominently enhance the effectiveness of anti-PD-1 immunotherapy. Spranger et al. have revealed that melanoma cell-intrinsic β-catenin signaling induces the exclusion of T-cells in the tumor microenvironment and renders resistance to anti-PD-L1/anti-CTLA-4 monoclonal antibody therapy [46]. Then, DeVito et al. proved that in autochthonous tumor models, pharmacologic suppression of Wnt ligand signaling can increase the efficacy of anti-PD-1 therapy by reversing dendritic cell tolerization and the recruitment of granulocytic myeloid-derived suppressor cells [47]. Nevertheless, these investigations mainly focused on the function of melanoma cell-intrinsic β-catenin signaling in surrounding immune cells in the tumor microenvironment. Whether tumorous Wnt/β-catenin signaling could affect the intrinsic characteristics of tumor cells and affect the execution of immunotherapy remains elusive. According to the results of our present study, the blockade of ferroptosis by systemic administration of liprostatin-1 might substantially ameliorate the influence of anti-PD-1 antibodies in melanoma, highlighting that the increased immunotherapy effectiveness induced by targeting tumorous Wnt/β-catenin signaling is largely dependent on ferroptosis. Previous reports combined with our present study indicate that tumorous Wnt/β-catenin signaling serves a versatile function in mediating the outcome of immunotherapy by simultaneously controlling tumor cell characteristics and tumor-infiltrating immune cells, which provides the molecular basis to support the great importance and promising translational potential of targeting Wnt/β-catenin signaling in melanoma immunotherapy. 

## 5. Conclusions

The present study reveals that Wnt/β-catenin signaling was prominently impaired in response to ferroptosis induction. Either pharmacological inhibition or the genetic knockdown of β-catenin could exacerbate ferroptosis via the regulation of MITF expression, with the downstream effector PGC1α and SCD1 greatly implicating it. What is more important is that pharmacological inhibition of either β-catenin or MITF could increase the treatment effect of anti-PD-1 antibodies in a preclinical mouse model by facilitating melanoma cell ferroptosis. Together, targeting the Wnt/β-catenin-MITF pathway might be an effective strategy for potentiating ferroptosis and enhancing the effectiveness of anti-PD-1 melanoma immunotherapy.

## Figures and Tables

**Figure 1 cells-11-03580-f001:**
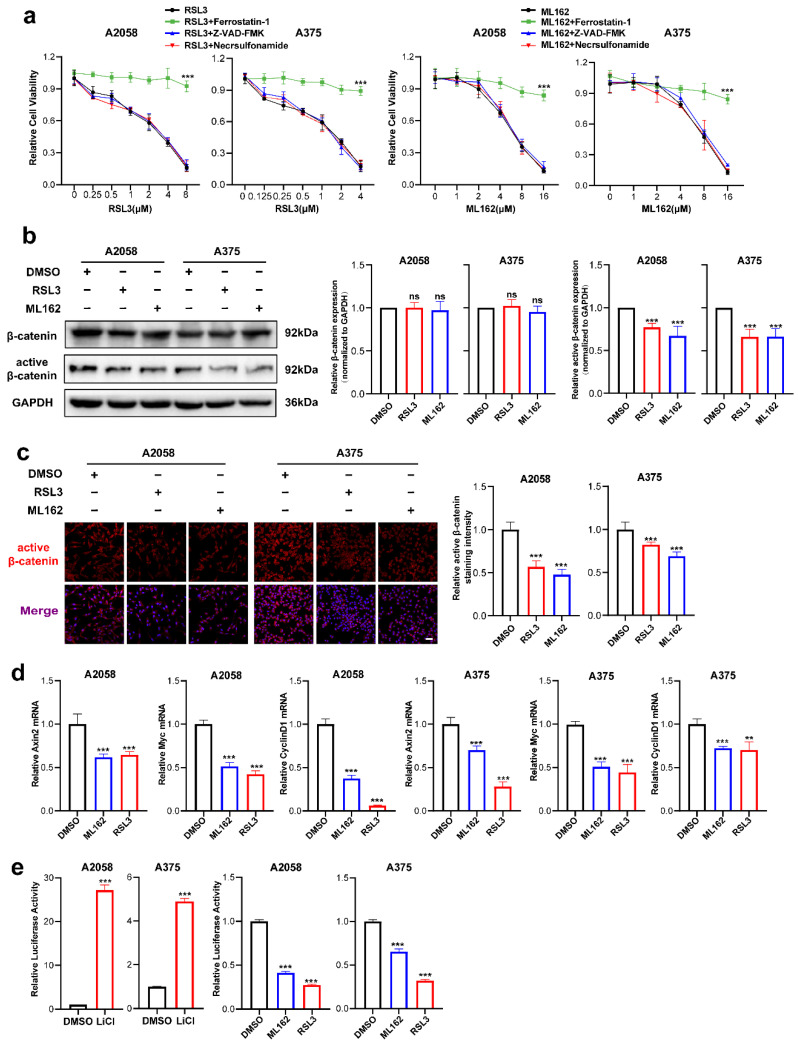
Wnt/β-catenin signaling is prominently suppressed in melanoma cell ferroptosis. (**a**) Cell viability of (0.125–8 μM) RSL3 or (1–16 μM) ML162-stimulated melanoma cells pretreated with 10 μM ferrostatin-1, 10 μM Z-VAD-FMK or 0.5 μM necrosulfonamide for 24 h. (**b**) Immunoblotting analysis of β-catenin and active β-catenin levels in response to either RSL3 or ML162 treatment in both A2058 and A375 melanoma cells for 24 h (RSL3, 2 μM for A2058 and 0.5 μM for A375. ML162, 4 μM for both A2058 and A375). (**c**) IF staining of active β-catenin in response to either RSL3 or ML162 treatment after treatment indicated in (**b**). Scale bars = 50 μm. (**d**) Relative mRNA levels of Axin2, c-Myc and CyclinD1 after treatment indicated in (**b**). (**e**) Transcriptional activity of β-catenin detected by Topflash reporter assay after the treatment with 20 mM LiCl, RSL3 and ML162 in both A2058 and A375 melanoma cells for 24 h (RSL3, 2 μM for A2058 and 0.5 μM for A375. ML162, 4 μM for both A2058 and A375). Data are presented as mean ± SEM of triplicates. *p*-value was computed using a two-tailed Student’s *t*-test. ** *p* < 0.01 and *** *p* < 0.001. ns, nonsignificant.

**Figure 2 cells-11-03580-f002:**
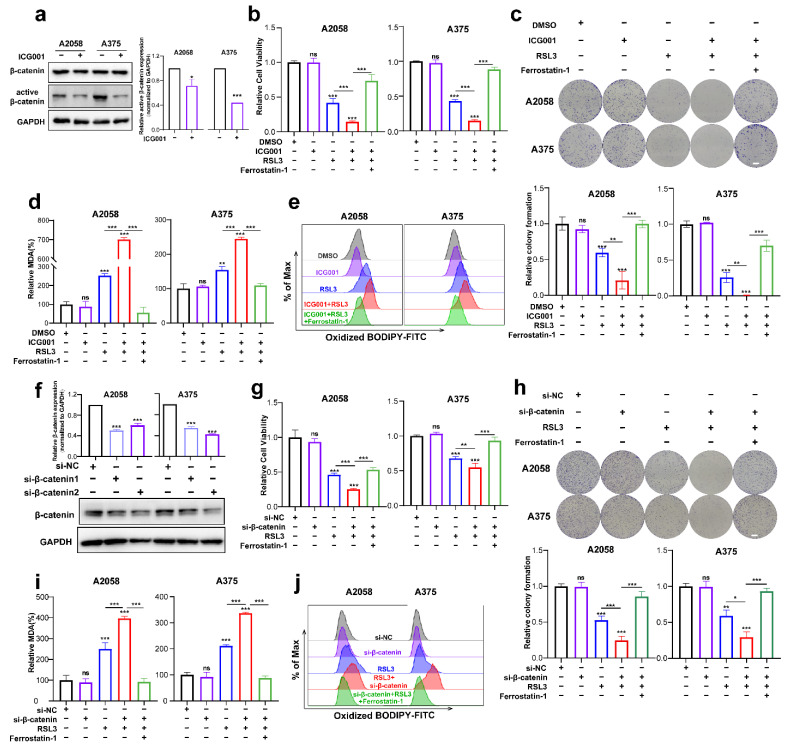
Targeting Wnt/β-catenin signaling exacerbates melanoma cell ferroptosis by promoting lipid peroxidation in vitro. (**a**) Immunoblotting analysis of β-catenin and active β-catenin levels in response to 6 μM ICG001 treatment for 24 h in both melanoma cell lines. (**b**,**c**) Relative cell viability and colony formation after the indicated treatment for 24 h in both melanoma cell lines (RSL3, 2 μM for A2058 and 0.5 μM for A375; 6 μM ICG001 and 10 μM Ferrostatin-1 for both A2058 and A375). Scale bars = 5 mm. (**d**) Intracellular MDA levels with the indicated treatment related to (**b**). (**e**) Lipid ROS levels of indicated cells were detected utilizing flow cytometry and employing C11-BODIP. (**f**) The knockdown efficiency of β-catenin in A2058 and A375 melanoma cell lines via immunoblotting analysis. (**g**,**h**) Relative cell viability and colony formation after the indicated treatment for 24 h in both melanoma cell lines with the knockdown of β-catenin (RSL3, 2 μM for A2058 and 0.5 μM for A375 and 10 μM Ferrostatin-1 for both A2058 and A375). Scale bars = 5 mm (**i**) Intracellular MDA levels with the indicated treatment related to (**g**). (**j**) By flow cytometry, lipid ROS levels in indicated cells were detected employing C11-BODIPY. Data were presented as mean ± SEM of triplicates. *p*-value was computed using a two-tailed Student’s *t*-test. * *p* < 0.05, ** *p* < 0.01 and *** *p* < 0.001. ns, nonsignificant.

**Figure 3 cells-11-03580-f003:**
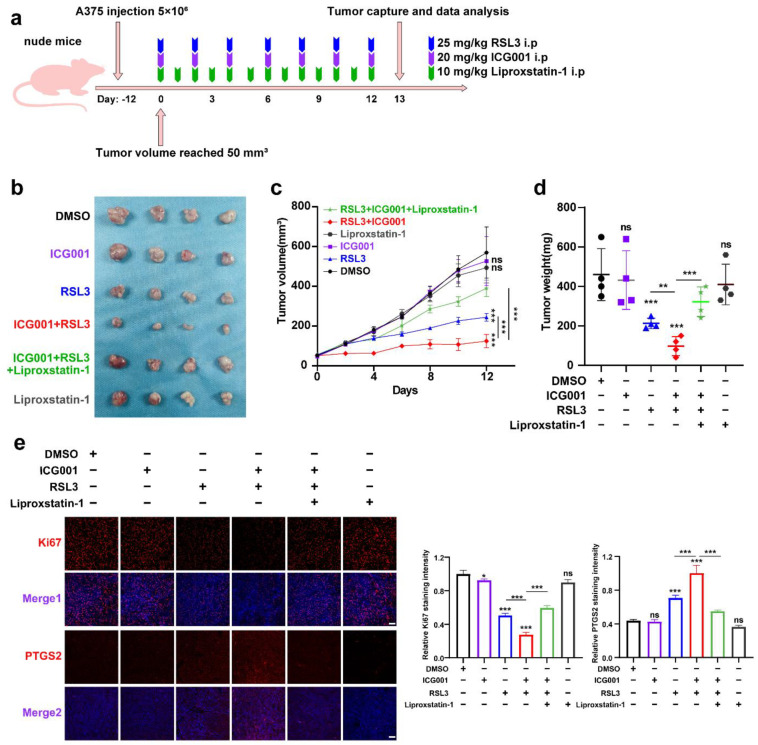
Targeting Wnt/β-catenin signaling exacerbates melanoma cell ferroptosis in vivo. (**a**) An illustration of the treatment strategy. (**b**–**d**) Captured images of isolated tumors in xenografted mice receiving the indicated treatments. Tumor volume and weight were calculated for each group, as shown in the right panel. (**e**) IF staining of PTGS2 and Ki67 in isolated xenograft tumors. Scale bar = 50 μm. Data are presented as mean ± SEM of triplicates. *p*-value was computed using a two-tailed Student’s *t*-test. * *p* < 0.05, ** *p* < 0.01 and *** *p* < 0.001. ns, nonsignificant.

**Figure 4 cells-11-03580-f004:**
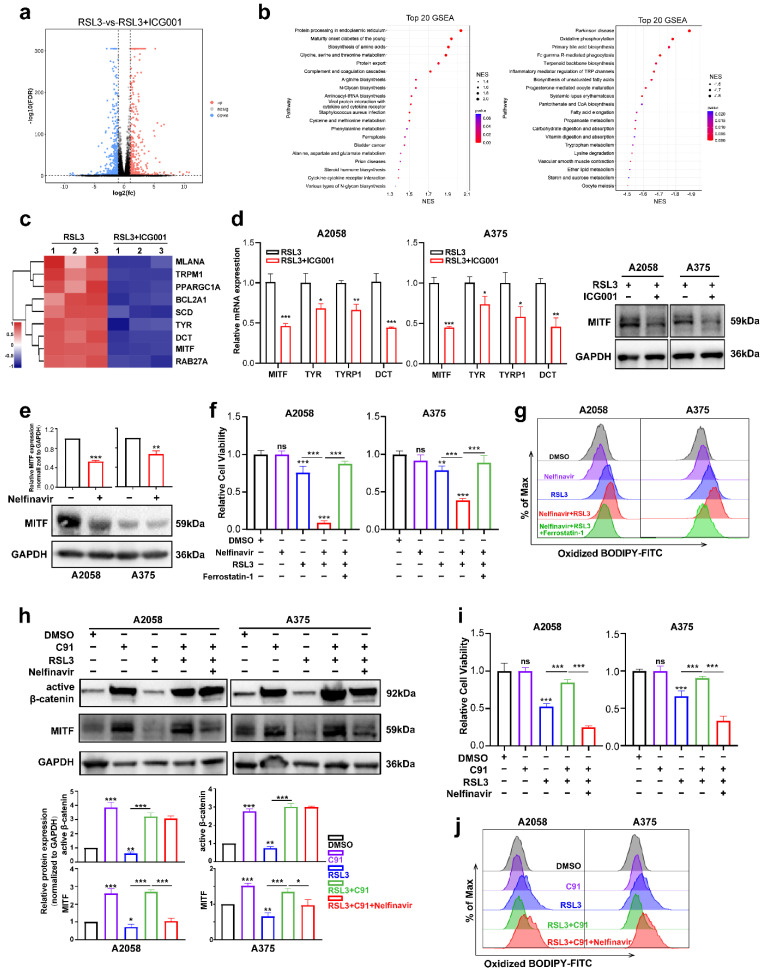
MITF mediates the influence of Wnt/β-catenin signaling on melanoma cell ferroptosis. (**a**) Volcano map displaying the differentially expressed genes between the RSL3 monotreatment group and the RSL3+ICG001 combined treatment group in the A2058 melanoma cell line. (**b**) The GSEA analysis displays the pathways enriched by differentially expressed molecules between the RSL3 monotreatment group and the RSL3+ICG001 combined treatment group. NES (normalized enrichment score). (**c**) Heatmap displaying that MITF and its downstream targets are differentially expressed between the RSL3 monotreatment group and RSL3+ICG001 combined treatment group. (**d**) Relative mRNA levels of MITF, TYR, TYRP1 and DCT, as well as protein levels of MITF, in the melanoma cell lines A2058 and A375, between the RSL3 monotreatment group and the RSL3+ICG001 combined treatment group. (**e**) Immunoblotting analysis of MITF after the treatment with 5 μM nelfinavir for 24 h in the melanoma cell lines A2058 and A375. (**f**) Relative cell viability after the indicated treatment for 24 h (RSL3, 2 μM for A2058 and 0.5 μM for A375; 5 μM nelfinavir and 10 μM ferrostatin-1 for both A2058 and A375). (**g**) Lipid ROS levels of indicated cells were detected utilizing flow cytometry and employing C11-BODIPY. (**h**) Immunoblotting analysis of active β-catenin and MITF after the indicated treatment for 24 h (RSL3, 2 μM for A2058 and 0.5 μM for A375; 5 μM nelfinavir and 5 μM C91 for both A2058 and A375). (**i**) Relative cell viability after the indicated treatment in the melanoma cell lines A2058 and A375. (**j**) Lipid ROS levels of indicated cells were measured utilizing flow cytometry and employing C11-BODIPY. Data are presented as mean ± SEM of triplicates. *p*-value was computed using a two-tailed Student’s *t*-test. * *p* < 0.05, ** *p* < 0.01 and *** *p* < 0.001. ns, nonsignificant.

**Figure 5 cells-11-03580-f005:**
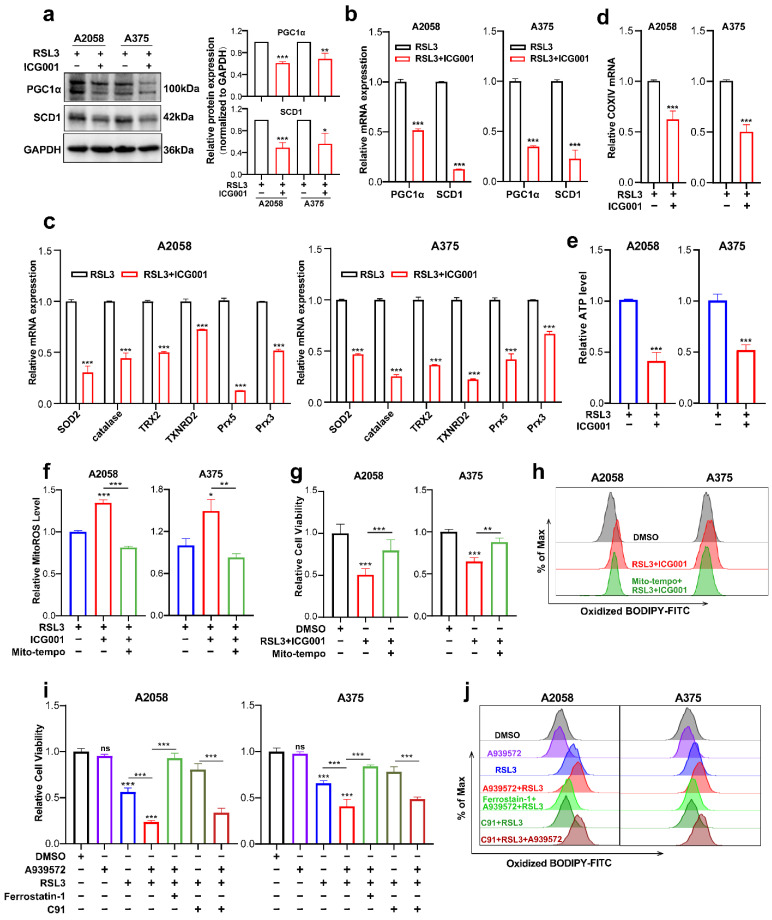
PGC1α and SCD1 are two main effectors downstream of the Wnt/β-catenin-MITF pathway in melanoma cell ferroptosis. (**a**,**b**) Immunoblotting and qRT-PCR analysis of PGC1α and SCD1 levels in the melanoma cells between the RSL3 monotreatment group and RSL3 plus ICG001 combined treatment group (RSL3, 2 μM for A2058 and 0.5 μM for A375 for 24 h. 6 μM ICG001 for both A2058 and A375 for 24 h). (**c**,**d**) qRT-PCR analysis of a panel of antioxidant enzymes and COXIV in the melanoma cells with the indicated treatment for 24 h (RSL3, 2 μM for A2058 and 0.5 μM for A375; 6 μM ICG001 for both A2058 and A375). (**e**,**f**) The level of mitoROS and intracellular ATP in A2058 and A375 with the indicated treatment for 24 h (RSL3, 2 μM for A2058 and 0.5 μM for A375; 6 μM ICG001 and 10 μM Mito-tempo for both A2058 and A375). (**g**) Relative cell viability of melanoma cell lines receiving the following treatment: RSL3, 2 μM for A2058 and 0.5 μM for A375, 6 μM ICG001 and 10 μM Mito-tempo for both A2058 and A375. (**h**) Lipid ROS levels of indicated cells were detected utilizing flow cytometry and employing C11-BODIPY which is related to (**g**). (**i**) Relative cell viability of both A2058 and A375 melanoma cell lines receiving a treatment with 2.5 μM A939572, RSL3 (2 μM for A2058 and 0.5 μM for A375), 10 μM ferrostatin-1 and 5 μM C91 for 24 h. (**j**) Lipid ROS levels of indicated cells measured utilizing flow cytometry and employing C11-BODIPY that is related to (**i**). Data are presented as mean ± SEM of triplicates. *p*-value was computed using a two-tailed Student’s *t*-test. * *p* < 0.05, ** *p* < 0.01 and *** *p* < 0.001. ns, nonsignificant.

**Figure 6 cells-11-03580-f006:**
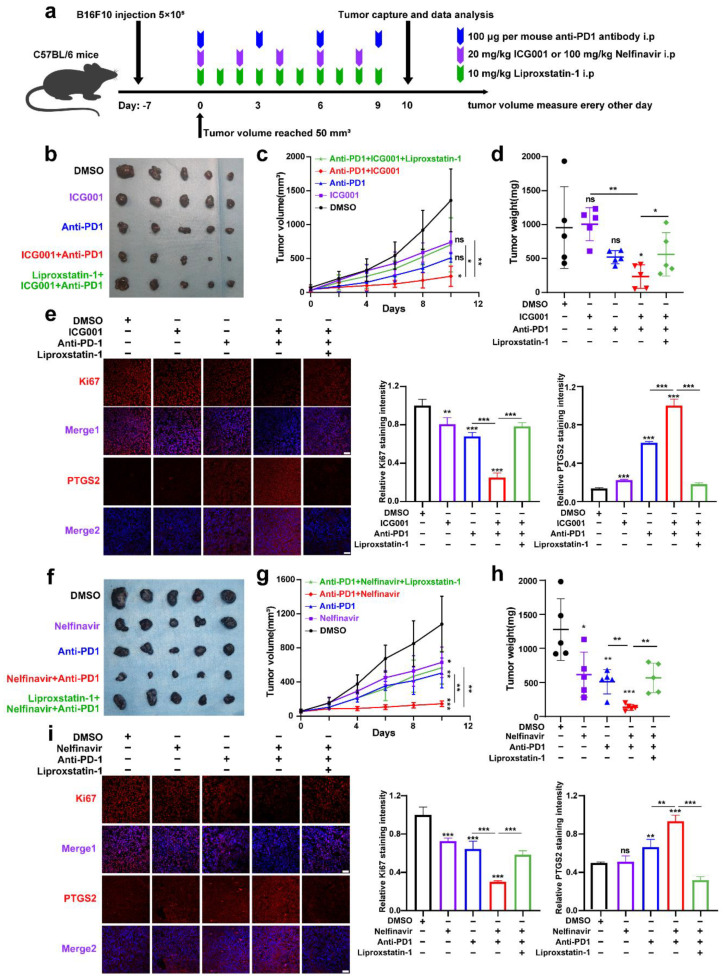
Targeting Wnt/β-catenin and MITF increases the effectiveness of anti-PD-1 antibodies partially via the induction of ferroptosis. (**a**) An illustration of the treatment strategy. (**b**–**d**) Pictures of isolated tumors from xenograft mice that received pharmacological inhibition of Wnt signaling and anti-PD-1 antibody treatment. Each group’s volumes and weights of tumors were computed and are shown in the right panel. (**e**) IF staining of PTGS2 and Ki-67 in isolated xenograft tumors with the indicated treatment related to (**b**). Scale bar = 50 μm. (**f**–**h**) Images of isolated tumors from xenograft mice that received pharmacological inhibition of MITF and anti-PD-1 antibody treatment. Each group’s volumes and weights of tumors were computed and are shown in the right panel. (**i**) IF staining of PTGS2 and Ki67 in isolated xenograft tumors with the indicated treatment related to (**f**). Scale bar = 50μm. Data were presented as mean ± SEM of triplicates. *p*-value was computed using a two-tailed Student’s *t*-test. * *p* < 0.05, ** *p* < 0.01 and *** *p* < 0.001. ns, nonsignificant.

**Figure 7 cells-11-03580-f007:**
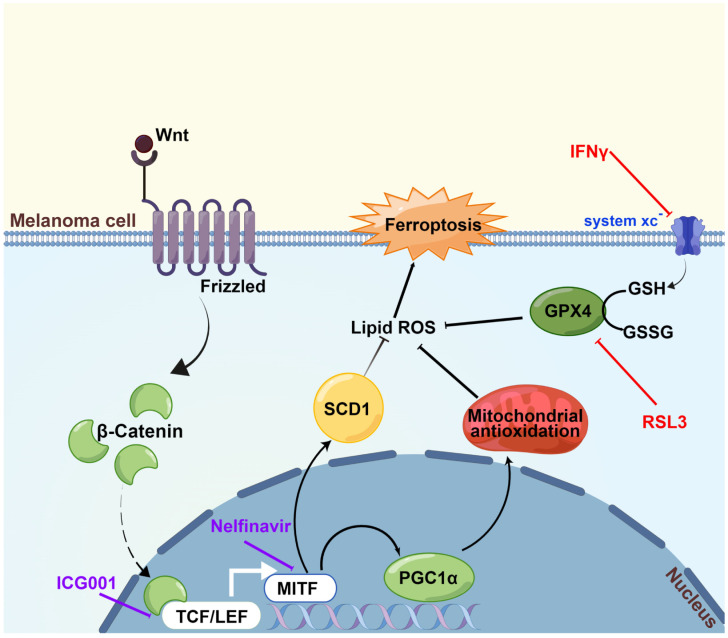
A proposed model of the function of the Wnt/β-catenin-MITF pathway in ferroptosis in melanoma. The activation of Wnt/β-catenin signaling can promote the transcription of MITF, leading to the upregulation of downstream PGC1α and SCD1, so that the generation of lipid peroxidation is prominently suppressed and ultimately ferroptosis is inhibited. Wnt/β-catenin signaling deficiency exacerbates ferroptosis in melanoma via the regulation of MITF. This figure was produced by Figdraw.

## Data Availability

The RNA-seq data utilized in current research has been deposited to the National Center for Biotechnology Information’s Sequence Read Archive (Sequence Read Archive study accession code PRJNA890052; https://www.ncbi.nlm.nih.gov/sra/PRJNA890052 (accessed on 13 October 2022)).

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
