# Peer review of "Targeting Wnt/β-Catenin Signaling Exacerbates Ferroptosis and Increases the Efficacy of Melanoma Immunotherapy via the Regulation of MITF"

_cells, 2022, doi:10.3390/cells11223580_

Round 1
Reviewer 1 Report
The manuscript entitles “Targeting Wnt/β-catenin signaling exacerbates ferroptosis and increases the efficacy of melanoma immunotherapy via the regulation of MITF” demonstrated that Wnt/β-catenin signaling deficiency exacerbates ferroptosis in melanoma via the regulation of MITF.
Although the significance of content, there are many recommendations for authors:
1. The introduction does not include all relevant references like discussion; the authors should add them.
2. 2.6 paragraph should be removed and added to paragraph 2.10.
3. In Materials and Methods authors did not talk about treatments; authors should be added a paragraph in which all treatments are described.
4. In 2.4 paragraph authors used the A2058 cell line to perform RNA sequencing; why the other cell line was not used?
5. In 2.11 and 2.12 paragraphs which cell lines were used? Please specify it.
6. In many paragraphs of Materials and Methods and Results the authors wrote “indicated treatments”. Authors should be added which treatment, treatment concentration and time-point of treatment.
7. In 239-240 lines, authors wrote: RSL3 and ML162 are two potent small-molecule inducers of ferroptosis by targeting 239 GPX4 activity [17, 18]. I think that it is better to discuss of this point in the introduction.
8. In paragraph 3.1 which treatment doses were used to perform Western Blot analysis and qPCR? Please specify it.
9. In paragraph 3.2 Authors tested ICG001: what is the timing of treatment? What is the treatment dose and timing of ferrostatin-1?
10. Authors should be describe how they choose the in vivo model and the treatment doses of the several compounds adding references.
11. The conclusions are not supported by the results; authors should be discuss the results even more during the discussion.
Author Response
- 1. The introduction does not include all relevant references like discussion; the authors should add them.
Response: Thanks. We have added some references to introduction part like discussion, and the revised contents have been marked with tracked changes and the added references have been marked with light.
- 2.6 paragraph should be removed and added to paragraph 2.10.
Response: Thanks for the reviewer’s kind suggestion. Actually, the antibodies used for both WB and IF were described in 2.6 paragraph, so it might be more appropriate to keep this part in 2.6 paragraph, instead of moving to paragraph 2.10.
- In Materials and Methods authors did not talk about treatments; authors should be added a paragraph in which all treatments are described.
Response: Thanks. Due to the complexity of the treatment conditions in different experiments, we supplied the information of the agents, concentrations and durations of indicated treatments in each figure legend instead of in Materials and Methods section, which can bring more help to make the experimental processes easier to be understood.
- 4. In 2.4 paragraph authors used the A2058 cell line to perform RNA sequencing; why the other cell line was not used?
Response: Thanks. Actually, it is a usual situation to perform RNA sequencing in one cell line and then conduct validation in other cell lines. Since that the ferroptosis induction effect of RSL3 combined with ICG001 is prominent in A2058 cell line and it is easier to culture A2058 compared to A375 cell line, we chose to use A2058 cell line to perform RNA sequencing. Similarly, we also used this strategy in our previous investigation published in Clinical Cancer Research (Guo et al. ATP-Citrate Lyase Epigenetically Potentiates Oxidative Phosphorylation to Promote Melanoma Growth and Adaptive Resistance to MAPK Inhibition. Clin Cancer Res. 2020 Jun 1;26(11):2725-2739).
- In 2.11 and 2.12 paragraphs which cell lines were used? Please specify it.
Response: Thanks. Both A2058 and A375 human melanoma cell lines were used in the experiments indicated in 2.11 and 2.12 paragraphs, and the relevant information has been added.
- In many paragraphs of Materials and Methods and Results the authors wrote “indicated treatments”. Authors should be added which treatment, treatment concentration and time-point of treatment.
Response: We feel sorry for previous obscure descriptions. We have supplied the information of the agents, concentrations and durations of indicated treatments in each figure legend, which can bring more help to make the experimental processes easier to be understood.
- In 239-240 lines, authors wrote: RSL3 and ML162 are two potent small-molecule inducers of ferroptosis by targeting GPX4 activity [17, 18]. I think that it is better to discuss of this point in the introduction.
Response: Thanks. According to the suggestion, we discuss the point in the introduction and relevant content has been revised as follows:
A cell death paradigm known as ferroptosis is characterized by iron-dependent lipid peroxidation [7], which is significantly involved in melanoma progression and associated with melanoma immunotherapy, and ferroptosis can be induced in vitro by some specific agents, with RSL3 and ML162 being as two potent small-molecule inducers of ferroptosis by targeting GPX4 activity [8, 9].
- In paragraph 3.1 which treatment doses were used to perform Western Blot analysis and qPCR? Please specify it.
Response: Thanks. The information of the agents, concentrations and durations of indicated treatments in both Western Blot and qPCR analysis has been added in the legend of Figure 1.
- In paragraph 3.2 Authors tested ICG001: what is the timing of treatment? What is the treatment dose and timing of ferrostatin-1?
Response: Thanks. The information of the agents, concentrations and durations of ICG001 treatment has been added in the legend of Figure 2.
- Authors should be describe how they choose the in vivomodel and the treatment doses of the several compounds adding references.
Response: Thanks. In order to testify the inhibitory effect of targeting wnt/β-catenin signaling on ferroptosis in vivo, a preclinical transplantation mouse model with the use of nude mice was established by subcutaneously injecting A375 melanoma cells. In addition, in order to figure out whether targeting wnt/β-catenin signaling or MITF could increase the efficacy of anti-PD-1 antibody, a xenograft tumor model was established with the use of immune-competent C57BL/6 mice by subcutaneously injecting B16F10 mouse melanoma cells. The relevant detailed description was added into paragraphs 2.13, 3.2 and 3.5, which is consistent with our previous published article in Journal of Investigative Dermatology (CAMKK2 Defines Ferroptosis Sensitivity of Melanoma Cells by Regulating AMPK‒NRF2 Pathway. J Invest Dermatol. 2022 Jan;142(1):189-200.e8.). In addition to this, the references of the treatment doses of the compounds Liproxstatin-1, RSL3, ICG001 and Nelfinavir were also added into the paragraph 2.13.
- The conclusions are not supported by the results; authors should be discuss the results even more during the discussion.
Response: Thanks for the reviewer’s suggestion. The conclusions section has been revised as follows:
The present study reveals that Wnt/β-catenin signaling was prominently impaired in response to ferroptosis induction. Either pharmacological inhibition or genetic knockdown of β-catenin could exacerbate ferroptosis via the regulation of MITF expression, with the downstream effector PGC1α and SCD1 greatly implicated in. What’s more important, pharmacological inhibition of either β-catenin or MITF could increase the treatment effect of anti-PD-1 antibody in pre-clinical mice model by facilitating melanoma cell ferroptosis. Together, targeting the Wnt/β-catenin-MITF pathway might be an effective strategy to potenti-ate ferroptosis and enhance the effectiveness of anti-PD-1 melanoma immunotherapy.

Reviewer 2 Report
This is very nice article describing the role of Wnt/β-catenin-MITF pathway in ferroptosis in melanoma. the introduction, methodology and results are well defined and originally written. the quality of figures is high, and all describe the actual findings reported in this study.
my minor comments are:
1- The authors could show statistical analysis for panel b and c in figure 1
2- The authors could show statistical analysis for panel a and f in figure 2
3- The authors could show statistical analysis for panel e in figure 3
4- The authors could show statistical analysis for panel e and h in figure 4
5- The authors could show statistical analysis for panel a in figure 5
6- The authors could show statistical analysis for panel e and I in figure 6
Author Response
1.The authors could show statistical analysis for panel b and c in figure 1
Response: Thanks for the suggestion. The statistical analysis for panel b and c in figure 1 has been added.
2.The authors could show statistical analysis for panel a and f in figure 2
Response: As suggested, the statistical analysis for panel b and c in figure 2 has been added.
3.The authors could show statistical analysis for panel e in figure 3.
Response: Thanks. The statistical analysis for panel e has been supplied.
4.The authors could show statistical analysis for panel e and h in figure 4
Response: Thanks. The statistical analysis for panel e and h has been supplied.
5.The authors could show statistical analysis for panel a in figure 5
Response: As suggested, the statistical analysis for panel a in figure 5 has been added.
6.The authors could show statistical analysis for panel e and i in figure 6
Response: As suggested, the statistical analysis for panel e and i has been supplied.

Round 2
Reviewer 1 Report
As already mentioned in the first report, I think that it is necessary to add all performed treatments in Materials and Methods. In this form, methods are not adequately described and results are not clearly presented.
Author Response
Comments and Suggestions for Authors:
As already mentioned in the first report, I think that it is necessary to add all performed treatments in Materials and Methods. In this form, methods are not adequately described and results are not clearly presented.
Response:
Thanks. According to the suggestion, we have added all performed treatments in Materials and Methods and relevant content has been revised as follows:
2.7. Lipid ROS production measurement
Onto 12-well plates, A2058 and A375 were plated at a density of 1.2 × 105 per well and cultured for 24 h to ensure cell adhesion. 1mL fresh medium mixed with following stimulations was then utilized to incubate the cells (RSL3, 2 μM for A2058 and 0.5 μM for A375; 6 μM ICG001; 5 μM Nelfinavir; 5 μM C91; 2.5 μM A939572; 10 μM Mito-tempo and 10 μM Ferrostatin-1 for both A2058 and A375). After 24 h, these cells were co-cultured with 1 ml of fresh DMEM comprising 5 μM C11-BODIPY 581/591 (cat. GC40165, GLPBIO) at 37℃ for 30 min. Subsequently, cells were subjected to twice wash with PBS, harvested with trypsinization, washed again, and kept in resuspension of 400 μl of PBS buffer containing 5% FBS. The cell suspension was analyzed on a flow cytometer to determine the amount of intracellular ROS [8, 9]. The maximum emission wavelength of oxidized BODIPY shifted from 591 nm to 510 nm, which indirectly reflected the level of lipid peroxide. The lipid ROS levels were examined employing the BD LSRFortessa Flow Cytometer (BD Biosciences, USA).
2.8. Malondialdehyde (MDA) assay
Utilizing an MDA assay kit (cat. A003-1-2, Nanjing Jiancheng, China), the relative MDA concentration in cell lysates was detected in accordance with the manufacturer’s guidelines. In brief, T75 flasks were used to seed melanoma cells (6×106 cells per flask) using stimulations related to Lipid ROS production measurement (2.7). Subsequent cells were trypsinized (cat. PB180228) and counted, and MDA then reacted with thiobarbituric acid (TBA) to produce an MDA-TBA adduct at 95°C under an acidic condition. At 532 nm, adduct absorbance was measured employing a Microplate Reader (Bio-Rad, Model 680, USA).
2.9. Mitochondrial ROS (mitoROS) detection
A2058 and A375 were co-cultured with following stimulations (RSL3, 2 μM for A2058 and 0.5 μM for A375; 6 μM ICG001 and 10 μM Mito-tempo for both A2058 and A375). In order to detect mitoROS, cells were harvested, washed once with DMEM, and suspended in 5 μM MitoSOX (M36008, Invitrogen) at 37°C for 15 min, keeping out of light. With pre-warmed PBS of 37°C, the cells were then washed twice and resuspended in pre-warmed PBS for detection by a flow cytometer (Beckman Coulter).
2.12. Measurement of ATP level and luciferase assay
In 6-well plates, melanoma cells A2058 and A375 were seeded at a density of 2×105 per well for 24 h. Cells were collected and lysed after following treatment for 24 h (RSL3, 2 μM for A2058 and 0.5 μM for A375; 6 μM ICG001 for both A2058 and A375). Then the ATP level was investigated using a mixture of 20 μL cell supernatant with 100 μL luciferase reagent employing an ATP Assay Kit (S0026, Beyotime). Luciferase activity, quantified the ATP level, was normalized to total protein analyzed utilizing the BCA Assay Kit. For canonical Wnt signaling activity luciferase assay, the SuperTOPFlash (D2505, Beyotime) was transfected and analyzed by using a luciferase reporter gene detection kit (RG005, Beyotime) using a luminometer (Promega GloMax 20/20) according to the manufacturer’s instructions.
2.13. Fe2+ assay
Cells were treated with following stimulations for 24 h (RSL3, 2 μM for A2058 and 0.5 μM for A375; 6 μM ICG001 for both A2058 and A375). The Fe2+ content in these cells was examined employing the Ferrous iron Colorimetric Assay Kit (E-BC-K773-M, elabscience, China) in accordance with the manufacturer’s guidelines. Spectrophotometry was adopted to measure the absorbance at a wavelength of 593 nm.
New paragraph:
2.14. Cell treatments
Firstly, to explore the ability of RSL3 and ML162 to induce ferroptosis using cell viability assay, (0.125–8 μM) RSL3 or (1–16 μM) ML162-stimulated A2058 and A375 melanoma cells were treated with 10 μM ferrostatin-1, 10 μM Z-VAD-FMK or 0.5 μM necrosulfonamide for 24 h. For western blotting, IF, Topflash and qRT-PCR assays, melanoma cells were treated with RSL3 or ML162 for 24 h (RSL3, 2 μM for A2058 and 0.5 μM for A375; ML162, 4 μM for both A2058 and A375). Moreover, to investigate the effect of Wnt/β-catenin signaling on melanoma cell ferroptosis, as well as the underlying mechanism, A2058 and A375 melanoma cells were treated with the following agents of indicated concentrations for 24 h (RSL3, 2 μM for A2058 and 0.5 μM for A375; 6 μM ICG001; 5 μM Nelfinavir; 5 μM C91; 2.5 μM A939572; 10 μM Mito-tempo and 10 μM Ferrostatin-1 for both A2058 and A375).
